Review  

**Subject Area:**
developmental biology

phylogenetic lineage reconstruction, dynamic barcoding, CRISPR/Cas9, scRNA-Seq, genetic switch

**Author for correspondence:**
Tzumin Lee
e-mail: leet@janelia.hhmi.org

# High-throughput dense reconstruction of cell lineages

Isabel Espinosa-Medina[1], Jorge Garcia-Marques[1], Connie Cepko[2] and Tzumin Lee[1]

[1]Howard Hughes Medical Institute, Janelia Research Campus, 19700 Helix Drive, Ashburn, VA 20147, USA
[2]Department of Genetics and Ophthalmology, Harvard Medical School, Boston, MA 02115, USA

TL, 0000-0003-0569-0111

The first meeting exclusively dedicated to the 'High-throughput dense reconstruction of cell lineages' took place at Janelia Research Campus (Howard Hughes Medical Institute) from 14 to 18 April 2019. Organized by Tzumin Lee, Connie Cepko, Jorge Garcia-Marques and Isabel Espinosa-Medina, this meeting echoed the recent eruption of new tools that allow the reconstruction of lineages based on the phylogenetic analysis of DNA mutations induced during development. Combined with single-cell RNA sequencing, these tools promise to solve the lineage of complex model organisms at single-cell resolution. Here, we compile the conference consensus on the technological and computational challenges emerging from the use of the new strategies, as well as potential solutions.

## 1. Introduction

Similar to family trees, all cells in any multicellular organism are connected by a genealogical line that relates every cell to the first single cell in the organism (zygote). Connecting the dots in this lineage tree, along with the identification of cell types, is as basic as finding the building blueprints of any organism. However, this fundamentally important task has been an ongoing challenge in most animals. We only know the cell lineage at single-cell resolution for a few organisms. The most remarkable one is *Caenorhabditis elegans*, which led Prof. Sulston to discoveries for which he shared the 2002 Nobel Prize in Physiology or Medicine.

Despite the existence of many strategies for lineage reconstruction, this meeting focused on the deployment of emerging tools built upon recent advances from studies of natural somatic mutations [1,2]. These tools use accumulated DNA mutations (CRISPR-induced in model organisms or natural mutations in human studies) to deduct cell lineage via phylogenetic analysis. This technology is powerful, in theory enabling the reconstruction of the lineage of any entire organism at single-cell resolution. The confluence of inexpensive DNA sequencing, generally applicable genome editing tools, and various strategies for the introduction of recording elements, led to the appearance of multiple studies aimed at this problem in a very narrow window of time. As Marshall Horwitz, one of the pioneers in the phylogenetic analysis of spontaneous mutations, pointed out: 'I think it's the sign of a promising idea that so many people have thought of it at once.' In addition to its importance to our understanding of developmental biology, reconstruction of cell lineages can have an impact in the clinic and in other applications, such as cancer therapy or tissue engineering. The advancement of this technology led to the choice of single-cell lineage reconstruction as a 2018 breakthrough of the year, by the journal *Science*.

### 1.1. The ideal tool for lineage tracing

Currently, we do not have a perfect method for lineage tracing and the gold standard will probably be different for each model organism. Yet we can

fantasize about what specific features the perfect tool should have. (i) Given that lineages are far less informative without knowledge of cell identities, the ideal tool should reveal both the lineages and the cell identities. Currently, the most powerful technology to characterize cell identity uses single-cell RNA sequencing (scRNA-Seq), suggesting that the ideal tool for lineage tracing be compatible with this approach. (ii) Cell state may rapidly change over the course of a single cell cycle. The ideal tool should therefore be able to record such changes. (iii) Most tools only reveal the final picture, the leaves in the lineage tree. We also need to understand the identity of cells that are present only transiently, including cells undergoing apoptosis and progenitor cells that rapidly change state. (iv) Functional analyses relating gene activity and cell lineage will be essential to understand molecular mechanisms involved in developmental processes. Ascribing mutant phenotypes to specific lineages will require spatial and morphological cell and lineage information. (v) Scalability is critical for achieving whole-organism lineage analysis. In many scenarios, we will need to reconstruct lineage information for thousands, if not millions, of cells. When the organism is accessible for imaging, lineage information can be recorded as images which are then stored in our computers for all to share. This very rich form of information can include much more than lineage data (e.g. cellular movements). A beautiful example of this was presented by Anastasios Pavlopoulos for *Parhyale*, a crustacean model [3]. The complete cell lineages of outgrowing *Parhyale* limbs were reconstructed from multi-dimensional and multi-terabyte light-sheet microscopy image datasets using open-source software for cell lineaging and tracking [4]. However, even in optically tractable animals, the large number of cells may often impair tissue-scale lineage tracing. In those cases, as well as in the absence of imaging accessibility (as occurs in most models in use today), we need to record lineage histories using other memory substrates. Currently, the most accessible is DNA, an ideal, compact vehicle for data storage. For this reason, it is the medium of choice for lineage information when the specimen is not accessible for real-time imaging. Indeed, strategies based on the phylogenetic analysis of DNA mutations seem to meet the requirement for scalability and single-cell resolution. As one can control the transcription of the DNA region undergoing mutations, these strategies are also compatible with most RNA-seq analyses. As discussed below, these strategies may be compatible with other approaches that preserve morphological and spatial information, as well as enable the acquisition of information concerning transient cells.

# 2. From natural mutations to CRISPR/Cas9 dynamic barcoding

## 2.1. Lineage tracing based on somatic mutations

Accurate sequencing of the entire genome of every single cell of an organism may reveal enough natural somatic mutations to reconstruct its lineage. This is the premise for pioneering retrospective lineage tracing methods based on phylogenetic analysis of natural mutations, the only valid approach currently available for human studies ([1,2] and figure 1*a*(i)). Despite recent technological advances, the cost of high-throughput single-cell whole genome sequencing remains prohibitive. That is the reason why several groups attending this meeting, interested in unravelling human lineage and cancer progression, focus their efforts on specific genomic regions which accumulate mutations, including microsatellites, CNLOH (copy neutral loss of heterozygosity) and CNVs (copy number variations). These genomic regions can be accessed without the need for single-cell whole genome amplification and have allowed the reconstruction of lineage trees from human and mouse samples [5–9].

Several challenges remain, as highlighted by Ehud Shapiro, including the need to combine lineage information with cell type, state and anatomical characterization, as well as validation of the inferred lineages in the absence of ground-truth data. To address the latter, *in silico* simulations in which a reference tree is generated have been used to evaluate various reconstruction algorithms [10]. More recently, the emergence of CRISPR/Cas9 dynamic barcoding in model organisms opened new opportunities to validate the various reconstruction systems as well (figure 1*b*(ii)). As Ehud Shapiro suggested, these technologies could be used not only to solve lineage in those organisms, but to cross validate the results from classical natural somatic mutation approaches and translate them to human lineage reconstruction.

## 2.2. Lineage tracing based on CRISPR/Cas9 mutations

Lineage reconstruction based on CRISPR/Cas9 accumulative editing relies on the use of synthetic or endogenous sequences which can be targeted by Cas9 and a specific set of gRNAs. The targets, specific sequences recognized by each gRNA, can be placed next to each other as an array or dispersed across the genome (table 1). The pairing of each gRNA with its corresponding target directs Cas9 to make double-stranded breaks which, after an error-prone repair mechanism, result in deletions or insertions (indels), creating a record of genetic events (barcodes) over time (figure 1*b*(ii)). In addition, synthetic transgenes can be transcribed, allowing simultaneous lineage reconstruction and cell-type characterization via RNA sequencing. These new lineage tracing approaches are often called 'dynamic lineage tracing' or 'dynamic barcoding' (see Glossary). The first reported example of dynamic barcoding was GESTALT (genome editing of synthetic target array for lineage tracing), in which transgenic zebrafish embryos carrying an array of 10 different targets (the barcode) were microinjected with Cas9 and a set of gRNAs at the one-cell stage. Thousands of mutations were recovered from adult dissected organs allowing correct proof-of-principle reconstruction of known lineage relationships during germ layer patterning and discovery of widespread clonal dominance [11].

Since the first implementation of this technology, many other strategies have emerged in a very short time, supporting CRISPR/Cas9 barcoding as a valid method for lineage tracing in several model organisms. These studies hold enormous promise for resolving complex lineages and have been reviewed elsewhere [21–23]. However, technical challenges remain which have prevented lineage reconstruction at single-cell resolution [16,24]. These challenges constituted the main focus of discussion of the meeting, along with the introduction of imaging tools which could be complementary to barcoding methods (figure 1 and table 1).

royalsocietypublishing.org/journal/rsob Open Biol. 9: 190229

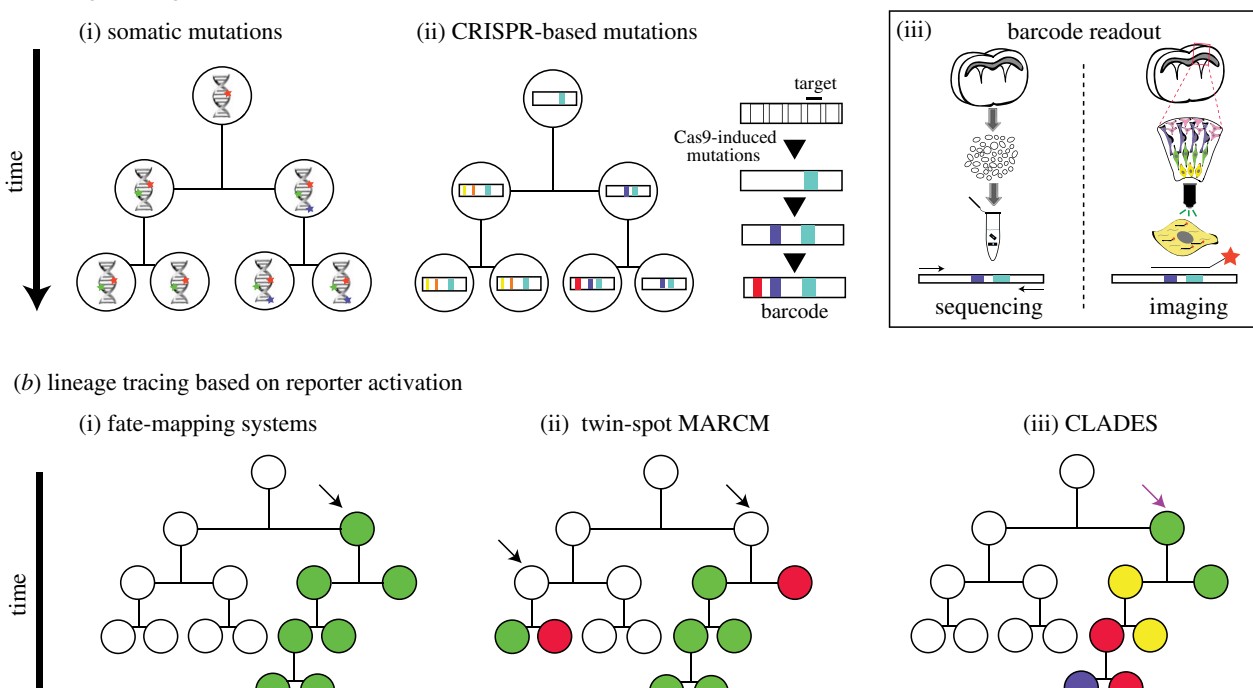

**Figure 1.** Summary of lineage tracing methods. (*a*) Based on cumulative mutations for retrospective phylogenetic reconstruction. (i) Somatic mutations accumulate naturally during development. (ii) Mutations induced by Cas9 accumulate on predefined targets along time. (iii) Barcode readout through tissue dissociation followed by single-cell isolation and sequencing from DNA or RNA (left) or through fluorescent *in situ* hybridization (orange star probe) of transcribed barcodes on intact tissue samples (right). (*b*) Based on reporter activation. (i) Conditional activation of a fluorophore in a progenitor cell (black arrow) and all its descendants (classically known as clonal labelling or fate-mapping). (ii) In twin-spot MARCM, induced interchromosomal recombination allows differential labelling of daughter cells derived from the same progenitor cell (black arrows). (iii) In CLADES, the induction by Cas9 (purple arrow) of a predefined cascade of fluorescent reporters in progenitor cells along time allows the distinction of the progeny for subsequent generations. This is a simplified representation of CLADES. Experimentally, the transitions from one colour to the next do not necessarily happen every cell division. For simplicity, all methods are exemplified using asymmetric lineages. In symmetric lineages and unlike other methods, CLADES could provide temporal information of emerging parallel lineages.

## 2.3. Technological challenges and possible solutions

### 2.3.1. Cell and barcode loss

A common problem encountered by many groups is low cell recovery from dissociated cell preparations made from tissues or entire organisms. As dissociated cells are often subjected to single-cell RNA-Seq (scRNA-Seq), their loss, which can be greater than 50%, can leave many gaps in the lineage reconstruction [25,26]. Among the many different techniques which have emerged that increase throughput, a new strategy called sci- (single-cell combinatorial indexing) was reported by Jay Shendure. Sci calls for running several rounds of split-and-pool of cells along with nucleic acid tagging [27,28]. Because each cell passes through a unique combination of wells, this results in unique transcript tagging. Although cell loss is still high using sci-, this strategy increases scalability exponentially while lowering total costs and has been applied to the reconstruction of developmental transcriptomic trajectories during mammalian organogenesis [29]. Similarly, the application of sci- to sort cells for dynamic lineage tracing would require combining many samples to compensate for the cell loss.

Another issue affecting CRISPR/Cas9-based lineage tracing is barcode loss, due to the reduced probability of sequencing a barcode in those cells that are captured. The reasons for this issue can be many. First, missing barcodes

in RNA-seq may result from low RNA expression levels. In this regard, Bushra Raj, Aaron McKenna and James Gagnon, who were pioneers in the GESTALT technology in the Schier and Shendure laboratories, relied on a heat-shock promoter to express the barcode. This system led to a low recovery rate (barcode information was recovered from approximately 6–28% of the total isolated cells [12]). They presented alternative promoters and regulatory sequences which increased expression levels both in zebrafish and *Drosophila* (B. Raj & A. McKenna 2019, unpublished results). Second, reduced transcript stability could stem from the presence of highly repetitive sequences and/or a high complexity of the transcript secondary structure, which could lead to degradation, contributing to barcode loss. Thus, placing repeated target sequences far away from each other, as well as using RNA-fold prediction software to avoid complex secondary structures, could help overcome this limitation [30]. At this point, it is worth noticing that single-cell DNA detection-based protocols have shown much higher barcode recovery rates (higher than 90%) than those based on RNA-seq. Anna Alemany from the Oudenaarden lab explained that targeted-DNA amplification is more efficient because barcode transcription might be tissue-specific, prone to silencing and scars might affect the half-life of the mRNA. This protocol requires more hands-on work and is lower throughput, as single-cell transcriptome libraries and targeted-DNA barcode libraries are generated independently [13].

**4**

**Table 1.** CRISPR-based barcode designs. From left to right, name and model organism, properties and schematic of various barcode designs available to date. Different targets are represented by different colours. Barcode properties: synthetic barcodes are transgenes artificially integrated into the host genome; endogenous barcodes are naturally present in the genomic DNA. Arrays involve multiple targets in tandem. Dispersed barcodes involve multiple targets present far apart in the genome. For each barcode, the number of identical (distinguishable or indistinguishable) or different targets is specified. Barcodes can be retrieved by sequencing (involves tissue dissociation, cell isolation and nucleic acid tagging) or by imaging (involves the hybridization of predefined fluorescent RNA probes to the edited or unedited barcode RNA). gRNAs are ubiquitously expressed in all systems except for MEMOIR, which relies on a Wnt-inducible gRNA. ID: molecular identifier.

| name (model) | ref. | barcode properties | scheme |
|---|---|---|---|
| GESTALT (*in vitro* and zebrafish) | [11,12] | synthetic array / 10 different targets / sequencing retrieval | |
| Scartrace (zebrafish) | [13] | synthetic array / 8 indistinguishable GFPs / sequencing retrieval | |
| LINNAEUS (*in vitro* and zebrafish) | [14] | synthetic dispersed / 32 indistinguishable RFPs / sequencing retrieval | |
| MARC1 (mouse) | [15] | synthetic dispersed / 60 different self-targeting gRNAs / sequencing retrieval | |
| SEQuoia (*Drosophila*) | [16] | synthetic array / 32 distinguishable variants of one target / sequencing retrieval | |
| molecular recorder (mouse) | [17] | synthetic array / 3 different targets / distinguishable (ID) multicopy integrations / sequencing retrieval | |

**Table 1.** (*Continued.*)

| name (model) | ref. | scheme | barcode properties |
|---|---|---|---|
| Cotterell & Sharpe (mammalian cells and zebrafish) | [18] |  | multiple endogenous arrays · 10 different targets/array · sequencing retrieval |
| Byungjin Hwang *et al.* (mammalian cells) | [19] |  | endogenous L1 repeats · >200 distinguishable targets · single gRNA/Cas9 base editor variant · sequencing retrieval |
| MEMOIR (mammalian cells) | [20] |  | synthetic array · 10 indistinguishable targets · distinguishable (ID) multicopy integrations · inducible gRNA · imaging retrieval (unedited versus edited state) |

royalsocietypublishing.org/journal/rsob    Open Biol. 9: 190229

Third, a standing limitation stems from the use of Cas9 nuclease to edit compact arrays of targets. Large deletions spanning several targets (inter-target deletions, see Glossary) and/or deletions that eliminate the primer-binding region impair total barcode recovery. In addition, the information content in the recovered sequences is reduced, and recorded information can be completely lost. Max Telford's group simulated these problems and showed that barcode loss through inter-target deletions has a major impact in lineage reconstruction accuracy [16]. Several alternatives to avoid long deletions were proposed at this conference. The placement of targets farther apart from each other [14,15] and/or the use of alternative Cas9 versions which introduce point mutations instead of indels [31] (see table 1 for more details on each of these barcode designs) are two such alternatives.

### 2.3.2. Barcode efficiency: editing frequency, target capacity and outcome variability

In addition to barcode recovery, other factors that impact the efficiency of CRISPR/Cas9-based lineage tracing concern the design of the target sequences and the properties of the editor. Design features, such as the number of available targets for editing (capacity) and the variability of editing outcomes, are important determinants of efficiency. As well, the editing frequency by the editor is an important factor [16,24]. Currently, the total number of distinguishable synthetic barcode targets is 10 in zebrafish [11,12], 32 in *Drosophila* [16] and 60 in mouse (see below) (although in the last two studies, not all targets were useful for lineage reconstruction). Several computational models presented at this meeting predicted that in order to reach single-cell resolution and scalability to thousands or even millions of cells, more than hundreds of targets would be required [16,24]. Moreover, these studies agree that controlling the editing frequency would expand the efficiency of recording over longer time frames and increase the accuracy of lineage reconstruction. Bushra Raj from the Schier lab presented scGESTALT (a combination of GESTALT lineage tracing and scRNA-Seq) and showed that controlling Cas9 expression, so that zebrafish barcodes were edited at two different developmental times, increased barcode variability over time. In this case, a higher temporal resolution facilitated the study of lineage relationships within a single-cell atlas of the zebrafish brain [12]. Using a different approach in *Drosophila*, Marco Grillo and Irepan Salvador-Martinez developed a synthetic barcode containing 32 variants of the same target sequence (each of them carrying one or two mismatches) integrated as an array in a single genomic locus. They demonstrated that introducing different target mismatches allows various editing frequencies in the same barcode, expanding its capacity to record over time [16].

In another study, Reza Kalhor reported the generation of transgenic mice bearing 60 independent integrations of self-targeting gRNAs (see Glossary) with various editing speeds, increasing barcode capacity by allowing evolution of the same sequences over time. Proof-of-principle experiments using this method demonstrated correct reconstruction of placenta, yolk sac and embryonic tissues, as the first lineages recorded in mice [15]. Unlike previous studies which used compact target arrays, this method avoids long inter-target deletions by placing the targets dispersed across the genome. Although the earliest mouse lineages could be tracked using self-targeting gRNAs which undergo fast inactivating mutations that cannot be overwritten, tracking later segregating lineages such as brain lineages required slower self-targeting gRNAs [15]. Those slower self-targeting gRNAs remain mutable over a longer time frame, which could confound lineage reconstruction at single-cell resolution.

In a different study called LINNAEUS, presented by Jean Philipp Junker, multiple dispersed copies of ubiquitously expressed RFP transgenes were targeted in zebrafish embryos, avoiding mutation over-writing and allowing barcode recovery by RNA-Seq. In combination with scRNA-Seq, LINNAEUS faithfully reconstructed known germline lineage relationships [14]. A limitation of this method, which could confound lineage inference, however, is that the multicopy targets are indistinguishable from each other. To address the latter, Michelle Chan from the Weissman lab presented a different version of these technologies for tracing mammalian embryogenesis. In this case, unique molecular identifiers (see Glossary) were introduced into independent transgenes carrying each unit of the transcribed barcode, consisting of an array of three targets, allowing distinction of mutations coming from different integrated copies. This lineage recorder recapitulated canonical mammalian tissue relationships and unveiled an endoderm population with extraembryonic origin [17].

### 2.3.3. Endogenous barcode sequences

All of the abovementioned studies were aimed at improving barcode efficiency, by regulating the editing frequency, increasing barcode recovery or increasing mutational variability. However, the low number of available targets remains the limiting factor to reach single-cell resolution when the lineages of thousands to millions of cells are to be reconstructed. A proposed solution by several groups was to target endogenous, instead of synthetic, sequences. This strategy avoids the current limitations of genetic engineering, which include low cargo capacity and difficulty in obtaining high copy number integrations. Endogenous target arrays suitable for lineage tracing have been identified in the zebrafish and mouse genomes by James Cotterell from the group of James Sharpe [18]. As in GESTALT, they rely on the use of Cas9 nuclease to edit compact arrays of targets, but interestingly in this case they found fewer inter-target deletions, possibly due to lower target sequence similarities. In another study, the group of Duhee Bang decided to target endogenous L1 repeats, present in thousands of copies, with a base editor variant of Cas9 which introduces point mutations instead of deletions or insertions [31]. They obtained ground-truth tree data by time-lapse imaging of HeLa cells transfected with a PiggyBac transposon carrying the Cas9 base editor and a gRNA. After four generations, they picked individual cells and sequenced the L1 repeats. Their study demonstrated accurate single-cell lineage reconstruction *in vitro* using this method [19]. However, the editing frequency of this Cas9 variant is low (0.06 edits per hour) and it remains to be determined if it is sufficient, and if barcode recovery from distant loci can be combined with single-cell transcriptomics, and scaled up for *in vivo* lineage tracing studies. More importantly, although targeting endogenous sequences has major advantages, as described above, we should be cautious and consider all possible deleterious effects on the genome [32,33].

## 2.4. Combining lineage and cell identity information

CRISPR/Cas9 barcoding has emerged as a powerful tool for lineage tracing. However, we still need to overcome the major technical challenges discussed above in order to reach single-cell resolution and scalability *in vivo*. Validating new lineage tracing approaches in the absence of ground truth would require proof-of-principle experiments using very well understood systems. Examples of fully reconstructed lineages include that of *C. elegans* and specific neuronal lineages in *Drosophila* [34,35], but reaching such a level of resolution in vertebrates has remained elusive. Nevertheless, there were several beautiful examples at this meeting of highly resolved lineage relationships obtained through continuous imaging and computational reconstruction, such as the process of neuromast regeneration presented by Hernan López-Schier, or the reconstruction of inner ear development in zebrafish by Cristina Pujades [36,37]. Thus, we could take advantage of those optically accessible and relatively small vertebrate systems to validate new lineage tracing methods.

To understand the lineage relationships among cells during development, lineage reconstruction with cell-type characterization is required. This would allow for the definition of the patterns that emerge as cell-type diversity is generated and allow for an appreciation of the lineage relationships among tissues and organs. Moreover, knowing both lineage and cell-type information would allow us to better assess inter-individual variability, essential to estimate the impact of lineage on developmental decisions.

At this point, it is worth noticing the fundamental conceptual difference, stated by Sean Megason in his talk, between 'cell lineage' and 'cell state manifold'. While 'cell lineage' refers to the topological structure that emerges from the connection of mothers with daughters through cell division, 'cell state manifold' defines a structure connecting changes in cell 'state' over time. Knowing only the branches of the lineage tree (which might vary in complex systems) or only the transcriptional cell states would not solve the question of how a single progenitor cell gives rise to the immense diversity of cells and tissues of an organism. In addition, one should be able to assess the effect of signalling events on lineage and cell-type formation to better understand developmental decisions.

Recent advancements in the field of scRNA-Seq have allowed high-throughput reconstruction of the transcriptomic landscape of several model organisms at several developmental stages [38–41]. Acquiring time series of scRNA-Seq data allows the generation of trajectories (named cell state manifolds above), which show how the transcriptome of cell populations changes over developmental time. Alex Schier highlighted the immense amount of relevant biological information within the developmental trajectories inferred from such data. As an example, he showed gene expression patterns linked to interesting cell biological changes during early notochord differentiation in zebrafish (A. Schier 2019, unpublished results). However, differentiation trajectories cannot be used alone to infer lineage relationships, as this can lead to erroneous interpretations [34]. Supporting the need to combine both cell lineage and state information, two examples at this conference including the study by Anna Alemany from the Oudenaarden lab and that by Michelle Chan from the Weissman lab, showed cell types with extremely similar transcriptomes but very different clonal origins [13,17].

Unravelling such differences could be the first step towards identifying similar cells with distinct functional properties.

To obtain even more accurate views, we should record transient states and signalling events, as well as lineage relationships, which can be retrieved from developing and terminally differentiated cells. Weixin Tang presented CAMERA (CRISPR-mediated analogue multi-event recording apparatus), a system in which external and internal cell signals induce the expression of a base editor Cas9 variant and gRNAs, which, in turn, mutate a safe locus which serves as a recorder. Endogenous pathways relevant during cancer development, the immune response or stress were recorded in human cells using this method [42] (W. Tang 2018, unpublished results). However, recording the number, order and complexity of signalling events and transcriptome transient states during development would require the capacity of this technology to be dramatically expanded. One way to do this could be to regulate the expression of multiple gRNAs under specific PolII promoters, which is possible if they are placed between self-cleaving ribozymes or tRNAs [43]. Similarly, one could record spatially regulated expression patterns, which would serve as 'genetic landmarks': spatial locators for single cells which would otherwise lose any anatomical reference after tissue dissociation. Finally, a prevalent topic at this meeting was the need for a cell cycle recorder, which would make for a major improvement in lineage reconstruction.

All the previously discussed recording methods rely on tissue dissociation prior to sequencing, but a full understanding of the function of biological systems also requires anatomical and morphological characterization of cells as tissues or organisms develop. Because of this, strategies to retrieve lineage and molecular information by imaging, while preserving tissue integrity, have also been developed.

# 3. From barcode sequencing to high-resolution imaging

## 3.1. Joint barcode and transcriptome retrieval by imaging

The first proof-of-principle method which combined CRISPR/Cas9 barcoding with *in situ* imaging readout, called MEMOIR, was developed by Michael Elowitz and Long Cai's labs. This system, which was tested in ES mouse cells, consists of multiple transcribed identifiers (see Glossary), each placed after a common array of 10 identical targets which 'collapse' when edited (deletions comprising one or more targets). Retrieval of the identifier along with one of the two possible mutational states (mutated or unmutated) is done by multiplexed single-molecule RNA fluorescence hybridization (smFISH) or seqFISH, uncovering both cell lineage and spatial information [20]. Although this technology allowed concurrent analysis of endogenous gene expression and cell lineage *in situ*, it suffered from limited resolution due to variable barcode expression. Increasing barcode diversity and expression level are therefore central to expanding the memory encoded in the MEMOIR system.

At this meeting, two laboratory members from the Elowitz lab presented upgrades of this technology to overcome those limitations. Ke-Huan K. Chow showed an improved and condensed MEMOIR 2.0 system based on integrases which avoids

double-strand breaks (DSBs) created by Cas9, demonstrating increased barcode variability for more accurate recording and lineage reconstruction *in vitro*, and which is being implemented *in vivo* using *Drosophila* as a model organism. Amjad Askary presented a novel *in situ* imaging method capable of reading more compact barcodes and distinguishing single-base edits with improved barcode recovery [44].

*In situ* methods for cell-type characterization based on RNA expression profiles have also been improved to reach higher detection efficiency and multiplexing capacity. Emma West, PhD student in Connie Cepko's lab, presented SABER (signal amplification by exchange reaction), a method which relies on the addition of long single-stranded DNA concatemers to the specific antisense probes and capable of amplifying the original FISH signal by up to 450-fold [45]. Its multiplexing capacity allowed *in situ* distinction of the 15 subtypes of retinal bipolar neurons, identified previously by classical methods and recent scRNA-Seq data [46]. Cellular morphology was revealed by wheat germ agglutinin (WGA) staining, and 3D reconstruction of morphology and quantification of FISH puncta allowed for the unambiguous assignment of cell identity within clones. Clonal marking was initiated using Cre to activate a fluorescent reporter.

Discovery-driven studies *in situ* would require full transcriptome profiling, but the high optical density of mRNAs in cells has remained a limiting step. Long Cai presented seqFISH+, a new implementation of the pre-existing technology (seqFISH) based on deterministic super-resolution microscopy which allowed imaging of up to 10 000 genes from single cells in mouse brain slices [47]. This study shows that while some gene expression is cell-specific, others define spatial regions but are characteristic of very different cell types. These spatial gene clusters were not resolved by scRNA-Seq, showing the advantage of conserving an intact tissue while performing molecular profiling of single cells. Also, this technology not only allows cell typing, but also distinguishes gene enrichment and subcellular RNA localization which can unveil interesting cell communication mechanisms.

Combined with lineage information obtained by the previously mentioned barcoding tools, SABER and seqFISH+ could uncover cell-type specific spatial and temporal relationships *in situ*. However, a standing limitation of the mentioned *in situ* readout tools is that they all require prior information about the target RNA sequences. Thus, distinguishing barcodes which accumulate random mutations such as GESTALT becomes challenging. An alternative would be to use other methods which have allowed *in situ* sequencing of unknown RNAs [48].

*In situ* barcode and transcriptome retrieval relies on RNA targeting, and the obtained images often contain empty spaces that lack resolution of cell morphology. Co-labelling by a cell-type agnostic stain (e.g. WGA) and immunofluorescence could help integrate cell-type and lineage information with morphological characterization within intact tissues [45].

## 3.2. High-throughput lineage tracing techniques based on reporters

Although alternative imaging-based strategies resolve lineages at a lower scale as compared to dynamic barcoding, they allow to conserve morphology and spatial information. These strategies for clonal analysis are typically based on

the conditional activation of fluorophores (genetic switches) in a particular progenitor cell, whose expression is then retained by all of its progeny. Techniques such as Brainbow expand the simultaneous clonal labelling capacity by incorporating multiple colour combinations [9,49,50]. However, this method cannot scale to resolve a great number of cell lineages, as clonally unrelated cells might be identically labelled. In addition, recombinases can result in the labelling of postmitotic cells, which can confound lineage interpretations [51]. An increase in scale can be achieved by viral infections with barcoded libraries. Libraries of greater than $10^6$ complexity can be used [52,53], but recovery of the barcodes relies upon cell dissociation with an approximate 50% loss in cells. Both of these methods of clonal marking will be improved if combined with an *in situ* method for the identification of cell type. The method discussed above by Amjad Askary may allow for the identification of cell type as well as barcode recovery *in situ*, using a lentiviral library to initiate clonal marking. The method uses phage RNA polymerases to amplify the barcodes in fixed tissue sections, followed by FISH detection of the resulting transcripts. Barcode recovery is thus not plagued by low RNA expression from the array *in vivo*. This method can be combined with FISH for marker genes to identify cell types in addition to barcodes.

To date, the only strategy which allows differential labelling of paired sister clones with high resolution is twin-spot MARCM (mosaic analysis with repressible cell markers) developed by Tzumin Lee in *Drosophila* [54] and later expanded to mouse as MADM (mosaic analysis with double markers) by Hui Zong at Liqun Luo's lab [55]. Both methods rely on the activation of two independent reporters after inducible interchromosomal recombination. In *Drosophila*, neuronal development is highly stereotypic and full neuronal lineages have been assembled by this method [35]. Also, this technology allows the characterization of mutant clones and has revealed the role of lineage-specific temporal factors on neuronal fate determination in the *Drosophila* central brain [56,57].

Songhai Shi presented his work using MADM to characterize neocortical gliogenesis in mice. This work revealed the precise timing of the transition between neurogenesis and gliogenesis, suggesting that both astrocytes and oligodendrocytes emerge from radial glial progenitors simultaneously, but, independently, in a quantal fashion (S. Shi 2019, unpublished results). Simon Hippenmeyer presented an alternative application of MADM: the induction of uniparental disomy to assess the consequences of genomic imprinting at single-cell resolution. This technique allowed him to demonstrate a requirement in parental imprinting for correct postnatal stem cell expansion of cortical astrocytes (S. Hippenmeyer 2018, unpublished results).

Despite MARCM and similar clonal labelling techniques being highly informative, they are limited by poor cell typing and a very low throughput which require the analysis of hundreds to thousands of samples in order to reach high resolution of lineages.

Two novel CRISPR strategies based on imaging presented by Jorge Garcia-Marques and Isabel Espinosa-Medina, members of the Tzumin Lee lab, promise to overcome those limitations. The first strategy, called CaSSA, involves the activation of fluorescent reporters by Cas9 and a gRNA through a conserved DNA mechanism known as single-strand annealing (SSA). SSA repairs DSBs occurring between two direct repeats by removing one of the repeats and also the intervening sequence [58]. Unlike repair by non-homologous

royalsocietypublishing.org/journal/rsob    Open Biol. **9**: 190229

end-joining, which creates random mutations (indels) at DSBs, SSA is scarless and predictable, allowing the reconstitution of reporter genes in a highly efficient manner. CaSSA facilitates access to cell types which require complex combinatorial labelling, by acting conceptually as an unlimited recombinase [59]. The second method, called CLADES, also takes advantage of SSA as repair mechanism, but in this case to activate a predefined cascade of gRNAs and fluorescent reporters [60]. By tracking the different markers expressed by the progeny, one can accurately reconstruct cell lineages within a single sample, unlike previous strategies which relied on lineage assembly from a large number of samples (see above). Both CaSSA and CLADES technologies are also being implemented in vertebrates (I. E. Medina 2019, unpublished results). Another powerful application of CLADES which could reach single-cell resolution would be the sequential editing of CRISPR barcodes. Computational simulations from a related study predicted a significant increase in accuracy of lineage reconstruction by combining predefined gRNA cascades with dynamic barcoding approaches [24]. Without combining this technology with dynamic barcoding, however, it is unclear how CLADES would perform resolving symmetric lineages in which the cascade progresses in all growing sublineages as cells increase exponentially.

# 4. Computational challenges for lineage reconstruction

## 4.1. Towards refined lineage reconstruction strategies

Sophisticated algorithms have been developed for constructing phylogenic trees. Despite general similarities, different assumptions apply to using CRISPR edits versus somatic mutations for lineage analysis. Mutable sites in the genome are innumerable, whereas Cas9 targets are usually very limited. Moreover, somatic mutations are transient—they can be overwritten by successive mutations. Conversely, CRISPR edits can be fixed, at least theoretically. These differences call for systematic evaluation of reconstruction algorithms for CRISPR-coded cell lineages, in addition to the optimization of Cas9 targets to augment coding capacity.

Given no ground truth for many lineages, simulations are being used for an assessment of the performance of recorders or tree-building methods for the reconstruction of CRISPR-encoded cell lineages. Max Telford noted that target dropouts following inter-target deletions could drastically reduce the accuracy of lineage reconstruction. Acknowledging this issue, Aaron McKenna and coworkers tried to improve GESTALT lineage reconstruction using inclusion of a penalized maximum-likelihood estimation [61]. Notably, the custom algorithm used in Michelle Chan's work in Jonathan Weissman's lab also involves searching for trees with highest likelihood [17]. However, such likelihood-based algorithms could be extremely slow when the tree-searching space is huge. Through benchmarking conventional phylogeny methods, Ken Sugino from Tzumin Lee's lab identified hierarchical clustering with the Russell-Rao metric and complete linkage as the best performing algorithm in the dense reconstruction of robustly encoded cell lineages without severe cell loss [62].

To date, however, no lineage reconstruction algorithms have been rigorously examined for their robustness across diverse recorders and various extents of cell/code dropout.

It remains unclear if any computer program can consistently reconstruct the underlying cell lineages to a high degree of accuracy, given the reality of the limits of actual experimental data. Reconstructed trees should carry some indication about the confidence level of any branch within the trees. To mobilize a larger community for establishing such optimal tree-building methods promptly, Jay Shendure promoted a DREAM challenge on this subject, which received support from meeting participants after an introduction by Pablo Meyer, a DREAM Challenge director.

Down the road, computer-assisted bioinformatic and imaging analyses are needed for (i) connecting partial trees in space and time to derive a complete tree with experimentally validated internal nodes, (ii) mapping the *in silico* reconstructed trees back to developing tissues to reveal the corresponding cell for each node *in vivo* and (iii) comparing trees across samples, genotypes and even species to unveil the cell lineage mechanisms of organism diversity (i.e. intra-species variations/ diseases and evolution). As hinted in many talks, including Sean Megason's illustration of cell state manifold and Jay Shendure's single-cell RNA-seq of organogenesis, synthesizing insightful cell lineage trees entails multi-modal information at both single-cell and genome-wide levels. Having a joint tree depository, potentially within the rapidly progressing cell atlas consortium, would expand the synergy into a larger community and greatly expedite the discovery of new biology.

## 4.2. Nomenclature clarification

To facilitate future communication, this meeting also sought clarification of terminologies characteristic of cell lineage reconstruction with cumulative CRISPR edits. First, each gRNA 'target' can be regarded as a basic 'unit' of the code. Second, the possible editing 'outcomes' in a gRNA target can be referred to as 'character states' or 'levels' of the unit. Third, the collective target states or unit levels in a cell constitute the 'barcode/scar/allele' of the cell. Regarding lineage reconstruction, there are two types of dropouts: 'capture dropout' contributes to cell loss, and 'collapse dropout' or 'excision dropout' typically refers to loss of targets/units due to inter-target deletion. As to basic lineage topology, we propose a broad adoption of the convention used by the *Drosophila* community to refer to various asymmetric-division lineages as Type 0/1/2 lineages [63]. Along this convention, we refer to symmetric-division lineages as Type 3 lineages (see Glossary).

# 5. Conclusion

At this meeting, many interesting talks and posters were presented which could not be covered in this review, as we decided to focus on the main technological challenges and emerging solutions leading to high-throughput reconstruction of cell lineages.

One thing that was clear is that inducible and dynamic barcoding technologies combined with single-cell typing and precise computational reconstruction algorithms have the potential to solve complete lineages in model organisms at single-cell resolution. The intersection of those strategies with imaging and functional studies seems fundamental to fully understand biological processes. Despite promising steps towards this goal, several technical challenges still need to be solved and will require a new generation of

royalsocietypublishing.org/journal/rsob    Open Biol. **9**: 190229

recorders and computational reconstruction methods. With regard to these limitations/challenges, several discussions of the goals of these technologies took place. One view is that we may not need full lineage reconstruction of a complex organism to greatly advance our understanding of, development. Robust methods for lineage analyses of selected tissues at defined times would provide most developmental biologists with the tools that they need to advance their studies, particularly in combination with perturbations of gene function. As there are different goals for different types of studies, it is worth considering the requirements for lineage data for each type of study.

As follow-up of this rapidly evolving field, another meeting named 'Hindsight' will take place at the Allen Institute in Seattle March 2020, organized by Jay Shendure and Michael Elowitz. This will also be an opportunity to discuss the progress within the DREAM challenge competition.

Finally, we would like to end this review by highlighting some comments from John Sulston, who described the lineage of *C. elegans*. Back in 2011, he commented on the importance of knowing what one is aiming for and noted the existence of power tools for lineage tracing. He was aware of one of our most complex challenges as biologists: to define our question and choose the right technology to solve it. Today, incredibly powerful tools are progressively making this dream come true.

Data accessibility. This article has no additional data.

Authors' contributions. All authors contribute to manuscript writing.

Competing interests. We declare we have no competing interests.

Funding. This Janelia meeting was administrated by Dr Janine Stevens and funded by the Howard Hughes Medical Institute.

Acknowledgements. We thank all the scientists who provided feedback on the sections citing their contribution at this meeting.

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

# Glossary

| | |
|---|---|
| *Barcode* | DNA recorder which accumulates lineage or other type of information in the form of mutations. Often composed of several targets in tandem (*array*) or *dispersed* across the genome. The barcode state in a cell is also called *scar* or *allele*. |
| *Barcode dropout/excision dropout/collapse* | Total or partial loss of the barcode sequence resulting from long inter-target deletions. |
| *CRISPR Cas9-based mutations* | Insertions, deletions or single-base substitutions at a target site resulting from the error-prone repair of a DNA break by Cas9 directed by a target-specific gRNA. |
| *Cell lineage* | The topological structure that emerges from the connection of mothers with daughters through cell division. |

| | |
|---|---|
| *Cell state manifold* | The topological structure connecting changes in cell 'state' over time often called *state trajectories*. Although the transcriptional profile of a cell is often used to define its 'state', other molecular or structural features could also serve as 'state' hallmarks. |
| *Dynamic lineage tracing* | vAccumulative DNA editing which records lineage information over time. Also called *dynamic barcoding* if the mutated sequences involve barcodes. |
| *Genetic switch* | A conditional sequence that can activate or inactivate the expression of an element in a system such as a fluorescent protein and an effector. |
| *Mutational outcomes/ levels/characters* | All the possible sequences resulting from an edition event on a target. |
| *Phylogenetic analysis of mutations* | Estimating the genealogical relationships of every cell within an organism by comparing their mutational profiles. |
| *Self-targeting gRNAs* | A gRNA which directs Cas9 to its own DNA encoding sequence. It contains a Cas9/gRNA recognition motif which makes that sequence a target for editing. |
| *Somatic or spontaneous mutations* | Genomic changes that occur naturally in cells. |
| *Target/unit* | A specific DNA sequence recognized by a gRNA antisense sequence necessary for Cas9 editing. |

| | |
|---|---|
| *Types of lineages* | Based on the *Drosophila* convention, lineage types are defined as: |
| Type 0 | Each division results in a single neuron and a progenitor. |
| Type 1 | The first division results in an intermediate cell (called GMC or ganglion mother cell in *Drosophila*) which divides again to generate two postmitotic cells. |
| Type 2 | The first division results in an intermediate progenitor cell (called INP in *Drosophila*) which undergoes four to six divisions to generate another INP and an intermediate cell that in turn generates two postmitotic cells. |
| Type 3 | Unlike Types 0, 1 and 2 which represent asymmetric expansion lineages, Type 3 defines a symmetric expansion lineage. Every division produces two identical daughter cells which continue to divide until their differentiation. |
| *UMI/ID.* | Unique molecular identifier (UMI) or identifier (ID) is an exclusive and short molecular 'tag' added to DNA sequences in order to distinguish them from each other. If added to synthetic barcodes, they allow distinction of multiple integrated copies. |